# Low Genetic Polymorphism in the Immunogenic Sequences of *Rhipicephalus microplus* Clade C

**DOI:** 10.3390/vaccines10111909

**Published:** 2022-11-11

**Authors:** Ismail Zeb, Mashal M. Almutairi, Abdulaziz Alouffi, Nabila Islam, Luís Fernando Parizi, Sher Zaman Safi, Tetsuya Tanaka, Itabajara da Silva Vaz, Abid Ali

**Affiliations:** 1Department of Zoology, Abdul Wali Khan University Mardan, Mardan 23200, Pakistan; 2Department of Pharmacology and Toxicology, College of Pharmacy, King Saud University, Riyadh 11451, Saudi Arabia; 3King Abdulaziz City for Science and Technology, Riyadh 12354, Saudi Arabia; 4Department of Chemistry, Abdul Wali Khan University Mardan, Mardan 23200, Pakistan; 5Centro de Biotecnologia and Faculdade de Veterinária, Universidade Federal do Rio Grande do Sul, Campus do Vale, Porto Alegre 91501-970, RS, Brazil; 6Faculty of Medicine, Bioscience and Nursing, MAHSA University, Jenjarom 42610, Malaysia; 7Laboratory of Infectious Diseases, Joint Faculty of Veterinary Medicine, Kagoshima University, 1-21-24 Korimoto, Kagoshima 890-0065, Japan

**Keywords:** *Rhipicephalus microplus*, immunogenic sequences, Pakistan

## Abstract

*Rhipicephalus microplus* tick highly affects the veterinary sector throughout the world. Different tick control methods have been adopted, and the identification of tick-derived highly immunogenic sequences for the development of an anti-tick vaccine has emerged as a successful alternate. This study aimed to characterize immunogenic sequences from *R. microplus* ticks prevalent in Pakistan. Ticks collected in the field were morphologically identified and subjected to DNA and RNA extraction. Ticks were molecularly identified based on the partial mitochondrial *cytochrome C oxidase subunit* (*cox*) sequence and screened for piroplasms (*Theileria/Babesia* spp.), *Rickettsia* spp., and *Anaplasma* spp. PCR-based pathogens-free *R. microplus*-derived cDNA was used for the amplification of full-length cysteine protease inhibitor (cystatin 2b), cathepsin L-like cysteine proteinase (cathepsin-L), glutathione S-transferase (GST), ferritin 1, 60S acidic ribosomal protein (P0), aquaporin 2, ATAQ, and *R. microplus* 05 antigen (Rm05Uy) coding sequences. The *cox* sequence revealed 100% identity with the nucleotide sequences of Pakistan’s formerly reported *R. microplus*, and full-length immunogenic sequences revealed maximum identities to the most similar sequences reported from India, China, Cuba, USA, Brazil, Egypt, Mexico, Israel, and Uruguay. Low nonsynonymous polymorphisms were observed in ATAQ (1.5%), cathepsin-L (0.6%), and aquaporin 2 (0.4%) sequences compared to the homologous sequences from Mexico, India, and the USA, respectively. Based on the *cox* sequence, *R. microplus* was phylogenetically assembled in clade C, which includes *R. microplus* from Pakistan, Myanmar, Malaysia, Thailand, Bangladesh, and India. In the phylogenetic trees, the cystatin 2b, cathepsin-L, ferritin 1, and aquaporin 2 sequences were clustered with the most similar available sequences of *R. microplus*, P0 with *R. microplus*, *R. sanguineus* and *R. haemaphysaloides*, and GST, ATAQ, and Rm05Uy with *R. microplus* and *R. annulatus*. This is the first report on the molecular characterization of clade C *R. microplus*-derived immunogenic sequences.

## 1. Introduction

Ticks and tick-borne pathogens have substantial economic effects on the veterinary and public sectors, mainly in tropical and subtropical countries where cattle populations have been addressed at risk of ticks and tick-borne diseases, representing a huge estimated economic impact [1]. To date, controlling ticks and tick-borne diseases remains a serious challenge [2]. Although chemical acaricides are used to control the tick infestation, their continuous and excessive use has led to the accumulation of residues in milk, meat, and in the environment. They also induce the development of acaricide-resistant tick populations [3,4]. Tick-derived protective antigens have been an alternative approach that have gained focus for the characterization of immunogenic sequences in different tick species [5,6].

Several immunogenic sequences have been identified in ticks in the last few years including cysteine protease inhibitor (cystatin 2b) [7], cathepsin L-like cysteine proteinase (cathepsin-L) [8], glutathione S-transferase (GST) [9], ferritin 1 [10,11], 60S acidic ribosomal protein (P0) [12], aquaporin 2 [13], ATAQ [14], and *R. microplus* 05 antigen (Rm05Uy) [15]. Based on their physiological importance, the potential anti-tick vaccine efficacy of these immunogenic proteins against multiple ticks has been determined [7,8,9,10,11,12,13], and Rm05Uy was suggested as a candidate antigen for inclusion in the future anti-tick vaccine development [15]. Ticks have undergone different evolutionary processes marked by morphological and genetic variations [16]. In particular, sequence polymorphisms in the Bm86, ATAQ, and cathepsin L-like cysteine proteinase genes in *Rhipicephalus microplus* strains have been reported in different geographical locations [17,18,19,20,21]. This phenomenon has been welcomed to characterize previously reported anti-tick vaccines in different geographical setups [6,20], and to determine their sequence homogeneity to be-assured before vaccination potential [18]. Reports have shown that polymorphisms in the tick-derived immunogenic sequences of different tick strains are associated with an impact on anti-tick vaccine efficacy [17,22,23,24].

In tropical and subtropical regions, *R. microplus* tick infestation is predominant [25]. Based on the *cytochrome C oxidase subunit* (*cox*), *R. microplus* species complex has been grouped into five distinct geographical clusters; clade A includes ticks from Africa, Asia, and South America, clade B includes ticks from southern China and northern India, clade C includes ticks from Pakistan, Myanmar, Malaysia, Bangladesh, and India, and *Rhipicephalus australis* and *Rhipicephalus annulatus* [25,26,27,28]. Pakistan is a subtropical country where more than 70% of the rural population is directly or indirectly dependent upon livestock and contribute approximately 60.07% and 11.53% to agriculture and gross domestic product values, respectively [28,29,30]. *Rhipicephalus microplus* infestations seriously affect livestock and cause substantial losses to the country’s economy [31,32,33,34]. Characterizing immunogenic sequences from *R. microplus* is necessary for future anti-tick vaccine development [35]. Indeed, systematic work is required to identify and investigate immunogenic sequences and subsequently use them for anti-tick vaccine development. This preliminary study aimed to characterize immunogenic sequences in *R. microplus* ticks collected from various geographical locations in Pakistan to infer their phylogenetic relationship and determine the sequence polymorphisms.

## 2. Materials and Methods

### 2.1. Study Area

Ticks were collected in five districts; Swat (35.2227° N, 72.4258° E), Shangla (34.8883° N, 72.6003° E), Upper Dir (35.3356° N, 72.0468° E), Lower Dir (35.3356° N, 72.0468° E), and Mardan (34.1989° N, 72.0231° E) located in Khyber Pakhtunkhwa (KP) (Northwestern geographical state of Pakistan previously known as North-West Frontier Province), Pakistan. Geographically, it is bounded to the West by Afghanistan, South-East by Punjab, South-West by Baluchistan, and Gilgit-Baltistan in the North. The KP province comprises a 101,741 km^2^ total area with varied elevation and climactic perspectives. The hilly regions are cold in winter and cool in summer, and the temperature markedly falls towards the North. The selected study area represents the main cattle-keeping agroecological zones with high tick burden (Figure 1).

### 2.2. Ethical Approval

Ethical consent was obtained from the advanced studies and research board of the Abdul Wali Khan University Mardan under Dir/A&R/AWKUM/2021/5466. Oral and written consents were obtained from cattle owners for tick collection.

### 2.3. Sample Collection and Morphological Identification

The partially engorged ticks collected in the field from cattle hosts were preserved in RNAlater^TM^ (Invitrogen, Carlsbad, CA, USA) and instantly shifted to the laboratory for morphological identification and molecular experimentation. Each tick sample was morphologically identified under a stereo zoom microscope (SZ61, Olympus Corporation, Tokyo, Japan) to the species level employing a previously published dichotomous key [36] during the first hour of collection. Global Positioning System took the geographical coordinates of each sample location for tagging the exact locations on the land cover map of KP province of Pakistan in ArcGIS V. 10.3.1 [37].

### 2.4. Nucleic Acid Extraction and cDNA Synthesis

Morphologically identified 20 *R. microplus* ticks representing each district were individually used as a sampling unit for molecular experimentation. Ticks were individually diced with a sterile scalpel blade in ice-cold phosphate buffer saline (pH 7.2). Subsequently, whole female tick tissues were separately taken in a single 1.5 mL tube and homogenized in a sterilized environment. The homogenized tissues were subsequently used in two series: genomic DNA extraction using DNA extraction Kit (Qiagen Ltd., West Sussex, UK) and RNA extraction using TRIzol^®^ Reagent/100 mg tissue (Ambion, Life Technologies, Carlsbad, CA, USA) following the manufacturer’s instruction. The extracted DNA and RNA were assessed for quantity and purity using a NanoDrop (Nano-Q, OPTIZEN, Daejeon, South Korea).

Prior to cDNA synthesis, the genomic DNA was removed by treating 1 µg/µL quantified RNA with 1 µL DNase I and RNase-free (1 U/µL), 1 µL DNase buffer, and 10 µL DEPC-treated water (Thermo Fisher Scientific, Inc., Waltham, MA, USA). The reaction was incubated at 37 °C for 30 min, followed by DNase I inactivation using 1 µL of 50 mM EDTA, and incubated at 60 °C for 10 min. Furthermore, 1 µg/µL RNA was mixed with 1 µL of 100 µM oligo (dT) and 10 µL DEPC-treated water and incubated at 65 °C for 5 min. Samples were chilled on ice for 1 min and subsequently pipetted with 4 µL first-strand reaction buffer (5×), 20 U/µL RiboLock RNase inhibitor, 2 µL dNTPs (10 mM), and 200 U/µL RevertAid M-MuLV RT (Thermo Fisher Scientific, Inc., Waltham, MA, USA). The reaction was incubated at 42 °C for 1 h followed by 70 °C for 5 min. The cDNA concentration and purity were determined using a NanoDrop (Nano-Q, OPTIZEN, Daejeon, Republic of Korea).

### 2.5. Primer Synthesis

Reference primers were used for the amplification of *cox* of ticks [38]. Tick-borne pathogens commonly occurring in this tick were screened by the amplification of partial mitochondrial *16S rRNA* for *Anaplasma* spp. (345 bp), and *gltA* for *Rickettsia* spp. (401 bp) as previously described [39,40]. A set of primers was used for the amplification of 897 bp fragment of the mitochondrial *18S rRNA* of piroplams (*Theileria*/*Babesia* spp.) based on the conserved regions of previously reported piroplasms *18S rRNA* mitochondrial sequences. In order to amplify the full-length Open Reading Frame (ORF) encoding immunogenic proteins, primers were designed based on the sequences retrieved from GenBank including cysteine protease inhibitor (cystatin 2b) (Accession numbers; KM588294, KC816580), cathepsin-L (JX502822–JX502830, MN175238–MN175239, KM272201–KM272202, KC707945–KC707946, AF227957), glutathione S-transferase (GST) (HQ337616–HQ337618, HQ337620, HQ337622–HQ337623, EF440186, AF077609), 60S acidic ribosomal protein (P0) (KC845304, KR697563, KP087926, EU048401), ferritin 1 (AY456681, AF467696, AY277902–AY277904), aquaporin 2 (KP406519), ATAQ (MF314445–MF314447, MG437296, MG437298, MG437299), and *R. microplus* 05 antigen (Rm05Uy) (KX611484, EF675686). All primers were examined to avoid the self-complementary hairpins, dimers, and difference in melting temperature using Vector NTI V. 11.5 (Invitrogen, Part of Life technologies, Carlsbad, CA, USA) (Table 1).

### 2.6. PCR Amplification

The *R. microplus*-derived DNA was amplified in a total volume of 25 μL PCR reaction containing a template DNA (50–100 ng/μL), 1X PCR buffer, 3 mM MgCl2, 0.2 mM dNTPs, 0.5 mM each forward and reverse primers, 1 U Taq DNA polymerase, and PCR water “nuclease free” (Thermo Fisher Scientific, Inc., Waltham, MA, USA). Thermal cycling conditions for *cox*, *Anaplasma* spp., and *Rickettsia* spp. were followed as previously described (Table 1), however, the condition of piroplams amplification consisted of an initial denaturation at 94 °C for 5 min, followed by 35 amplification cycles (94 °C for 30 s, 55 °C for 30 s, and 68 °C for 1 min), and a final extension step at 72 °C for 10 min.

In order to amplify the full-length ORF encoding immunogenic proteins, a template cDNA (~500 ng/μL) was used, and a similar PCR reaction was prepared as mentioned above. The initial denaturation was kept at 94 °C for 4 min followed by 35 cycles of denaturation at 94°C for 1 min, annealing at 60 °C (cystatin 2b, ferritin 1, ATAQ, and Rm05Uy), 57 °C (cathepsin-L), 54 °C (GST), and 50 °C (P0 and aquaporin 2) for 30 s, and extension at 72 °C for 1–2 min. The final extension was performed at 72 °C for 7–10 min, and then held at 4 °C until further processing. A negative control of PCR without cDNA and a positive control containing tick-specific actin primers were used for PCR validation [41]. All PCR reactions were performed in a PCR thermocycler (T100 Bio-Rad, Laboratories Inc., Hercules, CA, USA). PCR amplified products were resolved by electrophoresis on ethidium bromide-stained agarose gel (1.8%) and the results were visualized under UV light using a Gel Documentation system (UVP, BioDoc-It imaging system, UVP, LLC, Upland, CA, USA).

**Table 1 vaccines-10-01909-t001:** Primers used for ticks, pathogens, and full-length ORF encoding immunogenic proteins.

Organism/Gene	Primer Sequence	Tm °C, s	Amplicon Size	References
Ticks/*cox*	F: GGA ACAA TATA TTT AAT TTT TGG R: ATC TAT CCC TAC TGT AAA TAT ATG	55 °C, 60 s	801	[38]
Piroplasms (*Theileria*/*Babesia* spp.)/*18S rRNA*	F: ACC GTGCTAA TTGT AGGGCTA ATAC R: GAACCCAAAGACTTTGATTTCTCTC	55 °C, 30 s	897	This study
*Rickettsia* spp./*gltA*	F: GCAAGTATCGGTGAGGATGTAAT R: GCTTCCTTAAAATTCAATAAATCAGG	50 °C, 30 s	401	[39]
*Anaplasma* spp./*16S rRNA*	F: GGTACCYACAGAAGAAGTCC R: TGCA CTCA TCGT TTACAG	55 °C, 30 s	345	[40]
Tick’s full-length ORF coding genes				
Cysteine protease inhibitor (cystatin 2b)	F: ATGGCTTCTTTGAGAATCACCCCG R: TTAGGTAGATGTGCTGCTTCCTTCG	60 °C, 30 s	423	This study
Cathepsin L-like cysteine proteinase (cathepsin-L)	F: ATGCTTAGATTAAGCGTACTTTGCG R: TTAGACGAGBGGGTAGCTGGCCTG	57 °C, 30 s	999	This study
Glutathione S-transferase (GST)	F: ATGGCTCCTGTGCTCGGCTAC R: GCTTGTTTCATGGCTTCTTCTGC	54 °C, 30 s	672	This study
Ferritin 1	F: ATGTTTTGGTCGATGTTATGC R: CTAGTCTGACAGGGTCTCCTTGTCA	60 °C, 30 s	654	This study
60S acidic ribosomal protein (P0)	F: ATGGTCAGGGAGGAYAAGAC R: CTAGTCGAAGAGTCCGAAGCCCAT	50 °C, 30 s	957	This study
Aquaporin 2	F: AAT TCAGCAGC AGGAG AAGC R: CTGA TGCATA AAAAA CTCAG CAT	50 °C, 30 s	1043	This study
ATAQ	F: ATG GGAA GAATG AACA ACG AACGC R: TCAG GCCTC TTCCTC CGTTG GAAGC	60 °C, 30 s	1818	This study
*R. microplus* 05 antigen (Rm05Uy)	F: ATGGT GGCTT TCAAG GCAG CCC R: TTAA CCATGG GCCGG CGC ACCA	60 °C, 30 s	516	This study
Actin	F: GCATCCACGAGACCACG R: GGGGTGTAGAAGGAAGG	54 °C, 30 s	339	[41]

### 2.7. Purification, Cloning and Sequencing

The amplified PCR products were precipitated in 1 mL of 100% absolute ethanol and 40 μL of 3 M sodium acetate (pH 5.2) and kept at −20 °C for overnight incubation. The solution was purified with the GeneClean II Kit (Qbiogene, Carlsbad, CA, USA) and the amplicons were individually ligated to pGEM-T vector (Promega, Madison, WI, USA) according to the manufacturer’s instructions. The plasmid constructs were used to transform in *Escherichia coli* TOP 10 strain (Invitrogen, Carlsbad, CA, USA) using a thermic shock method and the resultant colony clones were screened with PCR employing the same primers modified with the addition of *Nde I* and *Hind III* restriction site sequences. All the obtained PCR positive products were bi-directionally sequenced (Macrogen Inc., Seoul, Republic of Korea).

### 2.8. Sequence and Phylogenetic Analysis

The obtained nucleotide sequences were analyzed in SeqMan V. 5.00 (DNASTAR Inc., Madison, WI, USA), and each consensus sequence from 100% identical sequences of *cox* and full-length ORF coding genes were subjected to BLAST (BLASTn for nucleotides *cox* sequence, and BLASTp and BLASTx for immunogenic sequences) analysis at NCBI. The homologous nucleotide (FASTA aligned) and protein (FASTA complete) sequences of closely related species were retrieved for downstream analysis [42]. The alignment and editing of sequences were performed in BioEdit sequence alignment editor V. 7.0.5 [43,44]. Furthermore, the phylogenetic analyses for *cox* nucleotide sequences and immunogenic proteins were individually constructed using the Maximum Likelihood (ML) and Neighbor-Joining (NJ) methods, respectively, in Molecular Evolutionary Genetics Analysis (MEGA-11) software [45]. The evolutionary distances were computed using the Poisson correction method. Each constructed tree comprises a branch support value (1000 bootstrap replicons) for nodes [45], and title at each taxon showing a GenBank accession number, tick species, and country. An outgroup sequences were taken for keeping the validity of inferred tree topologies. The obtained sequences were aligned pairwise, identity and nucleotide polymorphism were determined using DnaSP6 software V. 6.12.03 [46].

## 3. Results

### 3.1. Sequences Analysis

The full-length ORF sequences encoding cysteine protease inhibitor (cystatin 2b, 423 bp), cathepsin L-like cysteine proteinase (cathepsin-L, 999 bp), glutathione S-transferase (GST, 672 bp), ferritin 1 (654 bp), 60S acidic ribosomal protein (P0, 957 bp), aquaporin 2 (1043 bp), ATAQ (1818 bp), and *R. microplus* 05 antigen (Rm05Uy) (516 bp) of *R. microplus* were molecularly characterized. Prior to sequences characterization, *R. microplus* ticks were molecularly characterized and screened for pathogens including *Babesia*, *Theleiria*, *Reckettisa* spp., and *Anaplasma* spp. and only pathogens-free ticks were included. The obtained *cox* nucleotide sequence revealed 100% identity to the *R. microplus* reported from Pakistan. The deduced amino acid sequences showed maximum identities with the same species sequences available in the GenBank. All the obtained identical sequences for each gene were considered as a consensus sequence, and the nucleotide sequences were uploaded to GenBank under the accession numbers: OP379525 (*cox*), OP2119720 (cystatin 2b), OP2119714 (cathepsin-L), ON921299 (GST), OP312653 (ferritin 1), ON921298 (P0), OP312654 (aquaporin 2b), OP2119719 (ATAQ), and OP312655 (Rm05Uy).

### 3.2. Phylogenetic Analysis of cox

Based on the *cox* nucleotide sequence, the *R. microplus* was clustered in clade C together with the sequences reported from Pakistan, Myanmar, Malaysia, Thailand, Bangladesh, and India. (Figure 2).

### 3.3. Phylogenetic Analysis of Cystatin 2b

The cystatin 2b deduced amino acid sequence showed maximum identity with the same sequences of *R. microplus* reported from India (100%), Brazil (95.71%), and China (95%). The obtained cystatin 2b sequence showed identity between 48.53 and 92.14% with the cystatin sequences of *R. appendiculatus*, *R. sanguineus*, *R. haemaphysaloides*, *Dermacentor andersoni*, *D. silvarum*, *Haemaphysalis flava*, and *Ixodes persulcatus*. In the phylogenetic tree, the cystatin 2b sequence clustered with the most similar available sequences of *R. microplus* (Figure 3).

### 3.4. Phylogenetic Analysis of Cathepsin L-like Cysteine Proteinase

Cathepsin-L deduced amino acid sequence showed maximum identity (99.40–100%) with the sequences of *R. microplus* reported from India. The obtained Cathepsin-L sequence showed identity between 68.06 and 98.80% with the cathepsin-L sequences of *R. annulatus*, *R. haemaphysaloides*, *R. sanguineus*, *D. variabilis*, *D. andersoni, H. flava*, *H. longicornis*, and *I. scapularis*. In the phylogenetic tree, the cathepsin-L sequence was clustered with the most similar available sequences of *R. microplus* and *R. annulatus* in the GenBank (Figure 4).

### 3.5. Phylogenetic Analysis of Glutathione S-Transferase

The GST deduced amino acid sequence showed maximum identity with the same sequences of *R. microplus* from India (100%) and USA (99.55%), and *R. annulatus* from Egypt (100%). The obtained GST sequence showed identity between 87.44 and 98.65% with the GST sequences of *R. appendiculatus*, *R. sanguineus*, *D. silvarum*, *D. marginatus*, *D. variabilis*, *Amblyomma variegatum*, *H. longicornis*, and *I. scapularis*. In the phylogenetic tree, the GST amino acid sequence was clustered with the most similar available sequences of *R. microplus*, and *R. annulatus* (Figure 5).

### 3.6. Phylogenetic Analysis of Ferritin 1

Ferritin 1 amino acid sequence showed maximum identity with the same sequences of *R. microplus* reported from USA and India (100%), and China (99.42%). The obtained Ferritin 1 sequence showed identity between 85.88 and 96.51% with the ferritin 1 sequences of *R. Haemaphysaloides*, *R. sanguineus*, *D. silvarum*, *D. andersoni*, *D. variabilis*, *H. longicornis*, *H. flava*, *H. doenitzi*, *A. americanum*, *A. maculatum*, *I. scapularis*, *I. ricinus*, *Ornithodoros moubata, O. parkeri*, and *O. coriaceus.* In the phylogenetic tree, ferritin 1 was clustered with the most similar available sequences of *R. microplus* (Figure 6).

### 3.7. Phylogenetic Analysis of 60S Acidic Ribosomal Protein (P0)

The deduced amino acid sequence of P0 showed maximum identity with the same sequences of *R. microplus* from Cuba (100%), *R. sanguineus* from China (100%) and Cuba (99.85%), and *R. haemaphysaloides* from China (99.69%). The obtained P0 sequence showed identity between 91.85 and 99.37% with the P0 sequences of *D. andersoni*, *D. nitens*, *D. silvarum*, *A. cajennense*, *H. longicornis*, and *I. scapularis*. In the phylogenetic tree, the P0 sequence was clustered with the most similar available sequences of *R. microplus*, *R. sanguineus*, and *R. haemaphysaloides* (Figure 7).

### 3.8. Phylogenetic Analysis of Aquaporin 2

Aquaporin 2 deduced amino acid sequence showed maximum identity with the same sequences of *R. microplus* from USA (99.66%), and China (98.98%). The obtained aquaporin 2 sequence showed identity between 58.66 and 90.72% with the aquaporin sequences of *R. sanguineus*, *D. variabilis*, *D. silvarum*, *D. andersoni*, *I. scapularis*, and *H. qinghaiensis*. In the phylogenetic tree, the aquaporin 2 sequence was clustered with the most similar available sequences of *R. microplus* in the GenBank (Figure 8).

### 3.9. Phylogenetic Analysis of ATAQ

The ATAQ deduced amino acid sequence showed maximum identity with the same sequences of *R. microplus* reported from Mexico (97.85%), and *R. annulatus* from Israel (96.69%). The obtained ATAQ sequence showed identity between 70.69 and 94.38% with the ATAQ sequences of *R. decoloratus*, *R. evertsi*, *R. appendiculatus*, *Hyalomma marginatum*, and *D. variabilis*. In the phylogenetic tree, the ATAQ sequence was clustered with the most similar available sequences of *R. microplus* and *R. annulatus* (Figure 9).

### 3.10. Phylogenetic Analysis of Rm05Uy

The Rm05Uy deduced amino acid sequence showed maximum identity with the same sequences of *R. microplus* reported from Uruguay and China (100%), and *R. annulatus* from Egypt (100%). The obtained Rm05Uy sequence showed identity between 75.90 and 99.95% with the sequences of *R. sanguineus*, *D. andersoni*, *D. silvarum*, *H. flava*, *H. longicornis*, and *I. scapularis.* In the phylogenetic tree, the Rm05Uy sequence was clustered with the most similar available sequences (Figure 10).

### 3.11. Nucleotide Polymorphism

Cystatin 2b and GST sequences have shown no nucleotide polymorphisms in pairwise alignment with the homologous sequences reported from India. Six nucleotide polymorphisms were found in the cathepsin-L sequence corresponding to three nonsynonymous polymorphisms (0.6%). The P0 nucleotide sequence showed five nucleotide polymorphisms. Four nucleotide polymorphisms in each ferritin 1 and aquaporin 2 sequences were found. However, a single nonsynonymous polymorphism was found in aquaporin 2 (0.4%). Nucleotide polymorphisms in the ATAQ sequence were 28, as a result 13 nonsynonymous polymorphisms were found (1.5%). Two nucleotide polymorphisms were found in Rm05Uy (Table 2).

## 4. Discussion

Tick control is a priority for many countries in tropical and subtropical regions [47]. Their control largely depends on the repeated use of acaricides, and the identification and characterization of tick immunogenic sequences is necessary [4,15]. A systematic work on the molecular characterization of full-length open reading frame (ORF) encoding immunogenic proteins in the *R. microplus* ticks of Pakistan is not available. For this purpose, the driven approaches for the characterization of immunogenic sequences and analysis of their genetic homogeneity are necessary to pre-ensure the future development of anti-tick vaccines and their subsequent trials.

Studies have suggested the importance of *cytochrome C oxidases subunit* (*cox)* sequences in revealing the intra-species phylogenetic relationship complex of ticks [26,48]. In this study, the obtained sequences for *cox* revealed maximum identity with the formerly reported *R. microplus* sequence from Pakistan; phylogenetically belongs to clade C, and distantly placed from clade A of Brazil, Panama, and Cambodia, and clade B of China [26,27,28]. Screening of *R. microplus* for the detection of pathogens such as *Babesia*, *Theleiria*, *Rickettsia*, and *Anaplasma* species revealed negative results. It has been shown that the tick-pathogen interaction can influence the expression of tick genes and transcriptional shifts [49,50]. Therefore, PCR-based negative *R. microplus* were used in the amplification of full-length ORF sequences. Amplifications of tick-derived protein encoding sequences have been shown in several studies; cysteine protease inhibitor (cystatin 2b) [7], cathepsin L-like cysteine proteinase (cathepsin-L) [8], glutathione S-transferase (GST) [9], ferritin 1 [10,11], 60S acidic ribosomal protein (P0) [12], aquaporin 2 [51], ATAQ [14,21], and *R. microplus* 05 antigen (Rm05Uy) [15], and their variable anti-tick vaccine efficacies have been determined. The obtained sequences revealed maximum identities with the available sequences of *R. microplus* reported from different regions. The partial polypeptide P0 protein has been used as an anti-tick vaccine against both *R. sanguineus* and *R. microplus* infestations [12,52]. The reason was due to the highly conserved sequence of P0 protein among the *R. sanguineus*, and *R. microplus*. Cystatin 2b sequence from *R. microplus* revealed its evolutionary homogeneity with the sequences of same tick population from India, China, and Brazil and showed its proximal genetic association with *R. haemaphysaloides*. Similar results have been previously obtained [7,53,54], suggesting that cystatin sequences showed a maximum identity within *Rhipicephalus* species. The GST and cathepsin-L deduced amino acid sequences showed genetic resemblance between *R. microplus* and *R. annulatus*. The sequence identity for each *R. microplus*-derived cathepsin-L and GST with the sequences of *R. annulatus* were 99.80% and 100%, respectively. The sequences were highly conserved that might be the reason of the close association between the two tick species as previously reported [55,56,57,58].

The phylogenetic tree based on ATAQ sequences was similar to the previous findings [14,21]. Several ATAQ sequences of *R. microplus* revealed a high identity at the protein level (97.8–100%) that have been reported from Mexico and phylogenetically clustered within the same clade [21]. Ferritin 1 and aquaporin 2 deduced amino acid sequences were highly identical to the homologous sequences of *R. microplus* reported from India, USA and China. Ferritin 1 sequences retrieved from various tick isolates revealed that the sequence was highly conserved within tick genera and clustered together with the same sequences of their respective *Rhipicephalus* genera [10,11]. This high degree of intraspecific homology in ticks can be suitable for tick’s phylogenetic reconstruction [10,59]. Aquaporin 2 sequence shared the same clade with the aquaporin 2 sequences from USA [13]. This remarkable similarity among aquaporin 2 sequences in *R. microplus* supports their conserved homogeneity. The Rm05Uy deduced amino acid sequence was clustered with the most similar available sequence from Uruguay and China, and *R. annulatus* from Egypt. This analysis was consistent with the previous report [15]. The phylogenetic analysis uncovered that the tick’s local isolates were closely related to isolates from different countries with maximum protein sequence identities reported from India and China. A probable explanation for this could be the commercial cattle trade between the adjacent countries.

Nonsynonymous polymorphisms in ATAQ, cathepsin-L, and aquaporin 2 sequences were observed in this study. Nonsynonymous mutations in the immunogenic sequences of *R. microplus* populations have been reported in different geographical locations [17,18,19,20,21,57,60]. Genetic analysis provides evidence of gene function and genetic variation in ticks, the latter being suggested as a factor in the variable efficacy between tick species and geographical strains [17]. Evolutionary factors such as adaptation to the local hosts and environment can significantly change the nucleotide sequences and it is generally accepted that evolution proceeds toward greater complexity at both the organismal and genomic levels [61,62]. In Mexico, ATAQ protein sequences revealed a high degree of conservation between the Mexican *R. microplus* populations [21]. Similarly, the protein sequences of two different strains of *R. microplus* were also identical [15]. Complying with these results, low genetic variability was observed for the full-length ORF immunogenic sequences of *R. microplus*. In contrast, nonsynonymous polymorphisms of cathepsin-L were high among the *R. microplus* populations [8]. Identifying the level of genetic polymorphisms and phylogeny of *R. microplus* complex from a geographically distinct recognized clade C is imperative for effective anti-tick vaccine development [63].

The ultimate aim of antigen screening is to analyze the *R. microplus*-derived immunogenic sequences to decipher their hidden/conserved genetic features for an anti-tick vaccine against *R. microplus* complex species [21,64]. It is important to obtain genetic information from strains in different geographical regions to develop effective anti-tick vaccines against infestations by regional ticks [65,66]. To the best of our knowledge, there is no published report on the molecular characterization of full-length ORF immunogenic sequences in Pakistan. Thus, the current study ensured the *R. microplus*-derived sequences’ identity, phylogenetic association, and variability with several previously characterized immunogenic sequences and displayed a guide for the control of the Pakistani *R. microplus* strain. These sequences have been previously used as an anti-tick vaccine against different *R. microplus* strains [7,12,29,52,67], however, polymorphisms in the antigenic sequences may decrease the anti-tick vaccine efficacy in different geographical locations [17]. Future studies should aim to increase the current database of tick-derived full-length ORF immunogenic sequences that can then be evaluated for the development of an anti-tick vaccine.

## 5. Conclusions

This is the first report of the *R. microplus*-derived full-length ORF immunogenic sequences from Pakistan. Phylogenetic analysis of the deduced protein sequences described a close resemblance to the corresponding sequences of *R. microplus*, *R. annulatus*, *R. sanguineus,* and *R. haemaphysaloides*. The obtained *cox* sequence revealed the evolutionary assemblage of *R. microplus* in clade C, which includes ticks from Myanmar, Malaysia, Thailand, Bangladesh, and India. Furthermore, low nonsynonymous polymorphisms were found which may be considered for in anti-tick vaccine development against *R. microplus* clad C.

## Figures and Tables

**Figure 1 vaccines-10-01909-f001:**
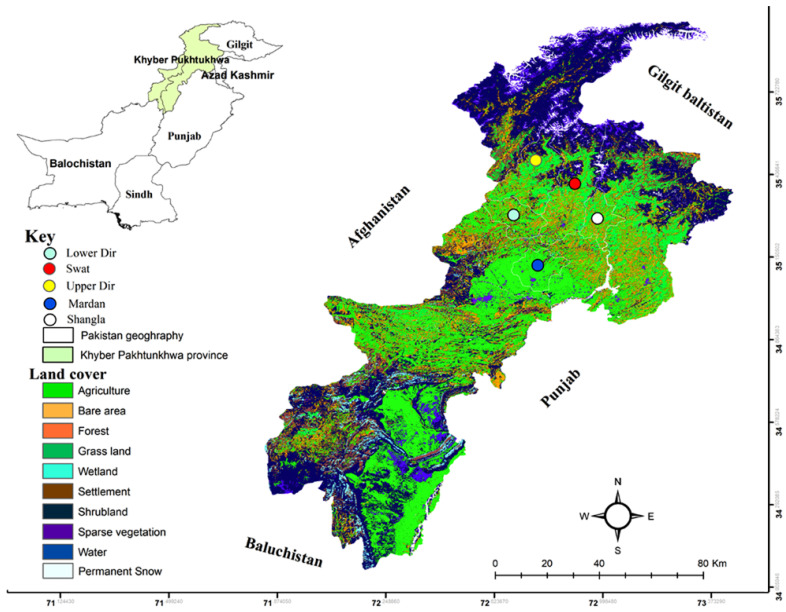
Land cover map of the study area representing the collection sites in different geographical locations in Khyber Pakhtunkhwa of Pakistan.

**Figure 2 vaccines-10-01909-f002:**
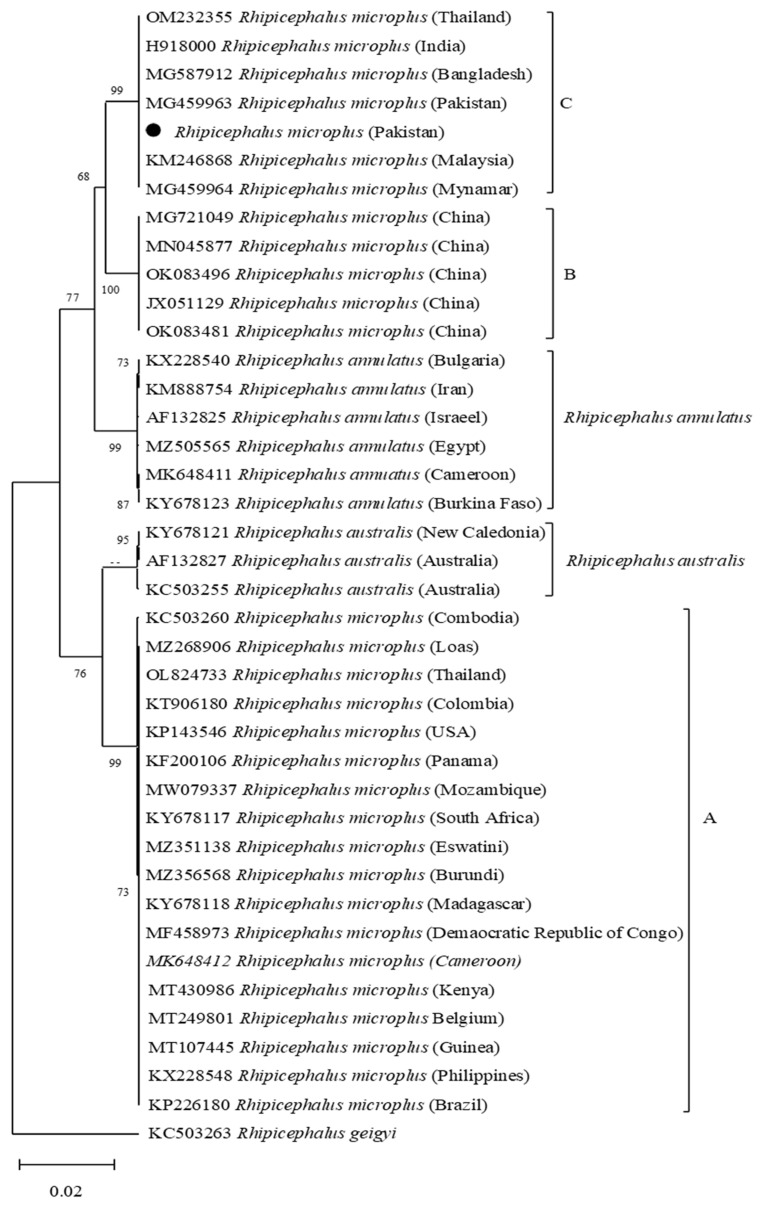
Maximum-Likelihood phylogenetic tree based on *cox* partial fragment of *Rhipicephalus microplus* and *Rhipicephalus geigyi* as an outgroup. The supporting values (1000 bootstraps) are indicated at each node, and the black circle represents the current study sequence.

**Figure 3 vaccines-10-01909-f003:**
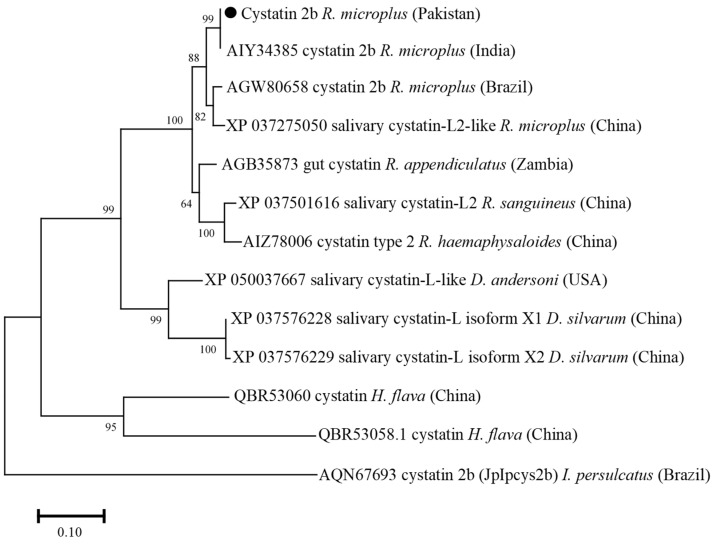
Phylogenetic tree based on the Neighbor-Joining method for the amino acid sequences of tick’s cystatin 2b, and *Ixodes persulcatus* as an outgroup. The supporting values (1000 bootstraps) are indicated at each node, and the black circle represents the current study sequence.

**Figure 4 vaccines-10-01909-f004:**
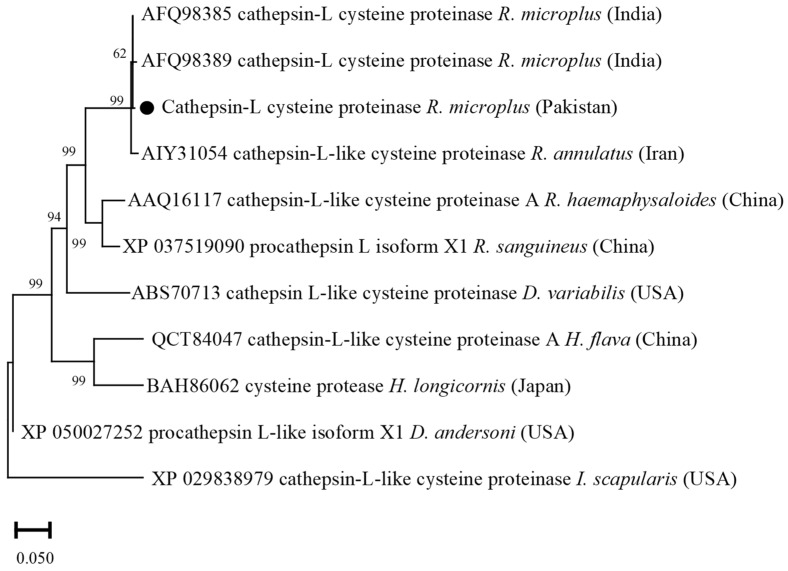
Phylogenetic tree based on the Neighbor-Joining method for the amino acid sequences of tick’s cathepsin L-like cysteine proteinase, and *Ixodes scapularis* as an outgroup. The supporting values (1000 bootstraps) for nodes are indicated, and the black circle represents the current study sequence.

**Figure 5 vaccines-10-01909-f005:**
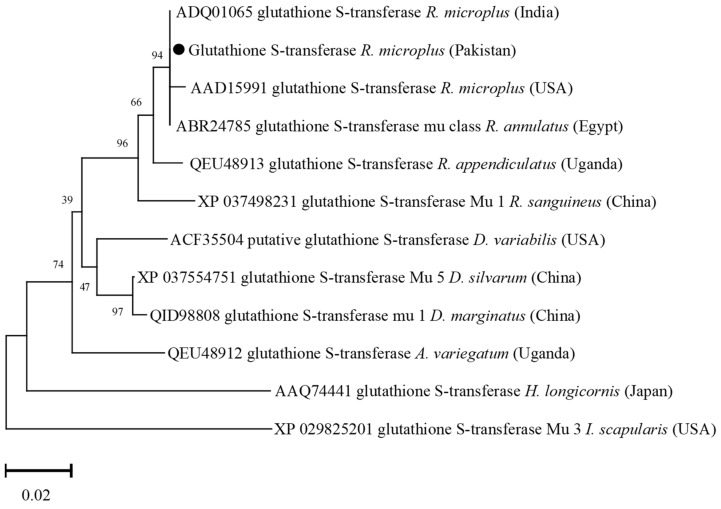
Phylogenetic tree based on the Neighbor-Joining method for the amino acid sequences of tick’s glutathione S-transferase, and *Ixodes scapularis* as an outgroup. The supporting values (1000 bootstraps) for nodes are indicated, and the black circle represents the current study sequence.

**Figure 6 vaccines-10-01909-f006:**
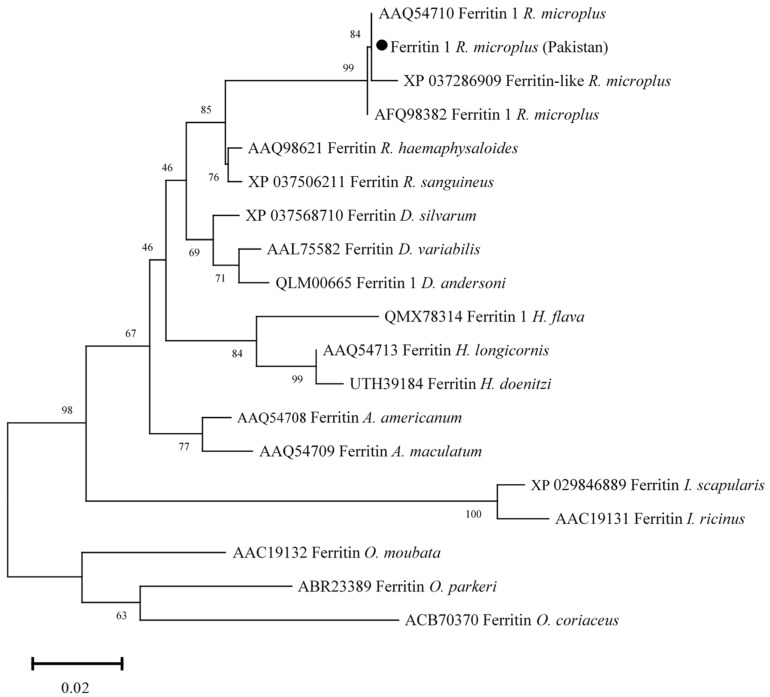
Phylogenetic tree based on the Neighbor-Joining method for the amino acid sequences of ferritin 1, and *Ornithodoros* spp. sequences were employed as an outgroup. The supporting values (1000 bootstraps) for nodes are indicated, and the black circle represents the current study sequence.

**Figure 7 vaccines-10-01909-f007:**
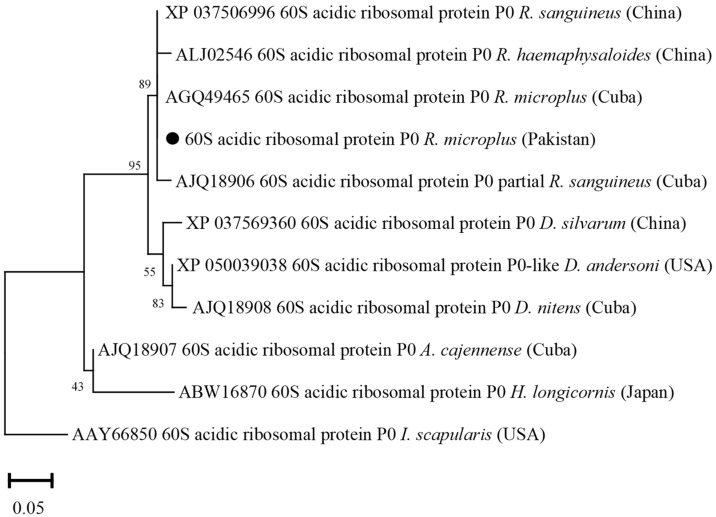
Phylogenetic tree based on the Neighbor-Joining method for the amino acid sequences of tick’s P0 and *Ixodes scapularis* sequence as an outgroup. The supporting values (1000 bootstraps) are indicated for nodes, and the black circle represents the current study sequence.

**Figure 8 vaccines-10-01909-f008:**
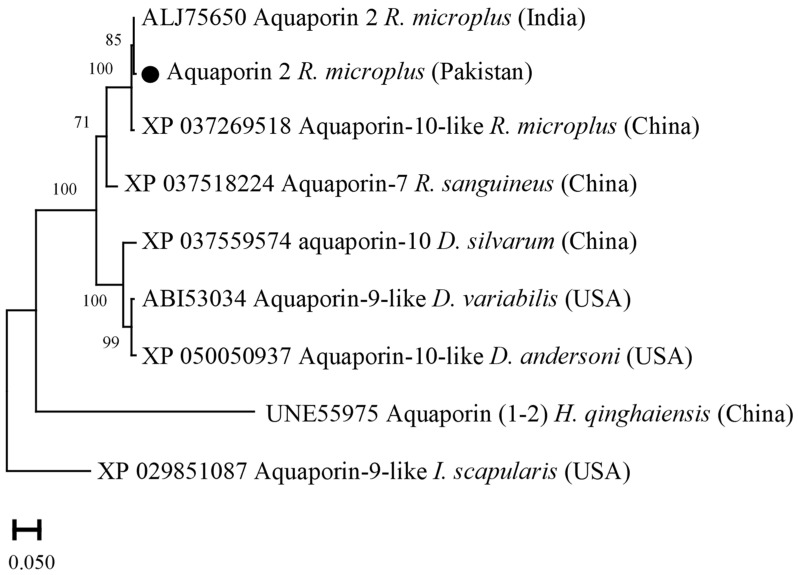
Phylogenetic tree based on the Neighbor-Joining method for the amino acid sequences of tick’s aquaporin 2, and *Ixodes scapularis* as an outgroup. The supporting values (1000 bootstraps) for nodes are indicated, and the current study sequence was represented with black circle.

**Figure 9 vaccines-10-01909-f009:**
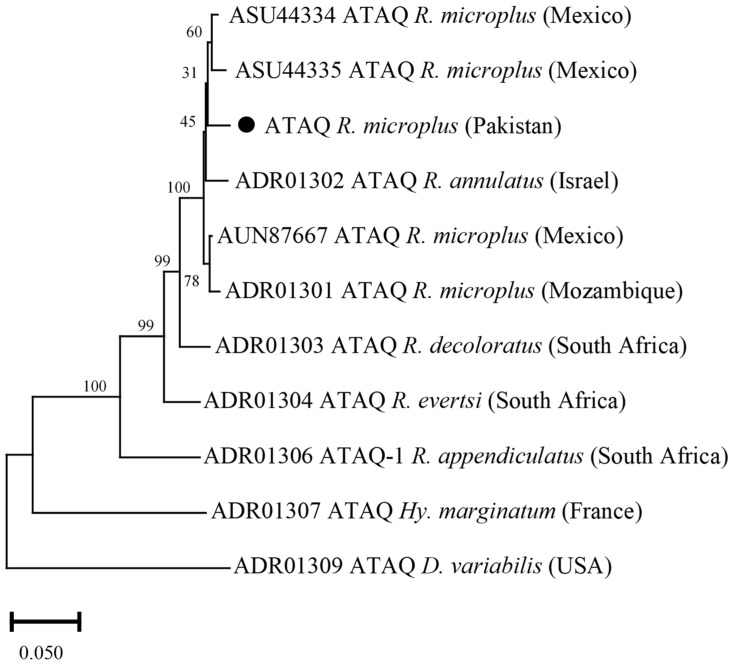
Phylogenetic tree based on the Neighbor-Joining method for amino acid sequence of tick’s ATAQ and *Dermacentor variabilis* as an outgroup. The supporting values (1000 bootstraps) are indicated for nodes, and the black circle represents the current study sequence.

**Figure 10 vaccines-10-01909-f010:**
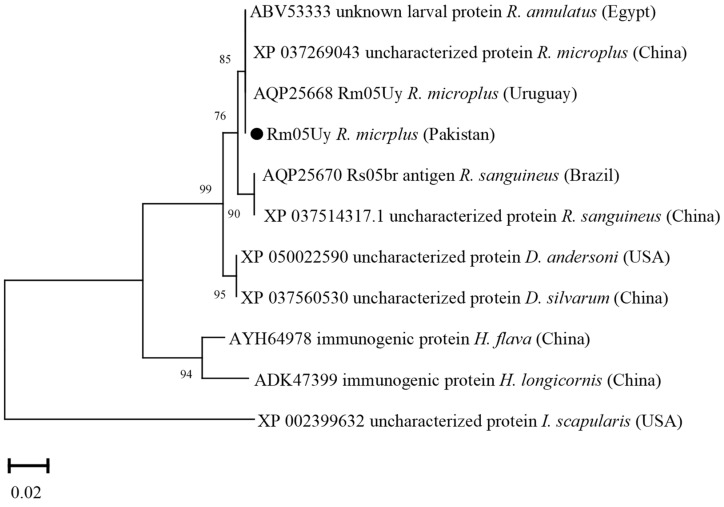
Phylogenetic tree based on the Neighbor-Joining method for the amino acid sequences of tick’s Rm05Uy and *Ixodes scapularis* as an outgroup. The supporting values (1000 bootstraps) for nodes are indicated, and the current study sequence was represented with black circle.

**Table 2 vaccines-10-01909-t002:** Identity, nucleotide polymorphisms, and their subsequent nonsynonymous polymorphisms of *Rhipicephalus microplus*-derived full-length ORF sequences compared to highly homologous published nucleotide sequences (GenBank) from different countries.

Gene	Country	Accession Number	Identity (%)	Polymorphic Nucleotides	Nonsynonymous Polymorphism
Cysteine protease inhibitor (cystatin 2b)	India	KM588294	100	-	-
Cathepsin L-like cysteine proteinase (cathepsin-L)	India	JX502822	99.40	T189C, A528C, A684G, A732G, T837C, T940A	Ser246Gly, Trp281Arg, Val315Asp
Glutathione S-transferase (GST)	India	HQ337620	100	-	-
Ferritin 1	USA	AY277902	99.39	G330A, A396G, T507C, T514C	-
60S acidic ribosomal protein (P0)	Cuba	KC845304	99.48	C75A, T151C, C195T, (T354C, A762G	-
Aquaporin 2	USA	KP406519	99.62	C781A, C873A, C961A, T963A	Leu254Ile,
ATAQ	Mexico	MG437296	98.46	G90A, A177G, G405A, C794A, C855G, A858G, A1001G, A1131G, G1137A, G1152A, C1170T, T1275C, A1284G, C1296A, C1318T, G1327A, G1415T, C1416T, T1483C, A1991C, A1562C, A1411C, G1640A, G1671A, C1695A, A1701G, T1710A, A1711C	Glu265Ala, Gln285His, Ile320Val, Gln334Arg, Pro440Ser, Val443Ile, Leu473Met, Cys495byArg, Glu497Asp, Asp521Ala, Gln537His, Ile567Met, Ile571Leu,
*R. microplus* 05 antigen (Rm05Uy)	Uruguay	KX611484	99.61	T207C, T319C	-

## Data Availability

The datasets presented in this study can be found in online repositories. The names of the repository/repositories and accession number(s) (OP379525, OP2119720, OP2119714, ON921299, OP312653, ON921298, OP312654, OP2119719, and OP312655) can be found in the article.

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
