# Peer review of "Low Genetic Polymorphism in the Immunogenic Sequences of Rhipicephalus microplus Clade C"

_vaccines, 2022, doi:10.3390/vaccines10111909_

Round 1
Reviewer 1 Report
The topic of this manuscript is scientifically relevant and it is in general well written. However, I would like to point that in the case of P0 sequences, the authors refers in the discussion section the unavailability of the R. microplus P0 protein sequence in the GenBank which is incorrect. The P0 sequence of R. microplus from a Cuban specimen is reported in the GenBank with the accession number KC845304 which is refered by the same authors on the Table 2 and appears in the paper Ticks and Tick-borne Diseases 6 (2015) 530–537. http://dx.doi.org/10.1016/j.ttbdis.2015.04.007. In this paper and also in the paper Vaccine 30 (2012), pp. 1782-1789 DOI information: 10.1016/j.vaccine.2012.01.011 is declared that the deduced 318 amino acid sequences from the amplified DNA sequences of the R. microplus and R. sanguineus ticks were 100% identical. It must be corrected in the phylogengenetic analysis and consequently in the discussion.
Author Response
Thank you for spotting this. The phylogenetic tree was re-constructed and the sentences were modified in the abstract, result, and discussion sections as suggested. In Table 2, the nucleotide sequence (KC845304) described the nucleotide polymorphism and associated nonsynonymous polymorphism in comparison with the current study 60S acidic ribosomal protein (P0) sequence, however phylogenetic tree was constructed using amino acid sequences.
Reviewer 2 Report
The manuscript describes the analysis of immunogenic sequences of Rhipicephalus microplus ticks from Pakistan. This study aims to characterize these immunogenic sequences and is presented as the first preliminary report from R. microplus phylogenetic clade C from this region, with impact in the development of an anti-tick vaccine.
The manuscript data could be soundness it correctly presented, since it provides new data (and regarding the importance of tick control and related tick-borne diseases transmission). However, there are several important faults: data that is not clear, coherent, and/or is incorrectly presented. The results presentation should have been more carefully addressed during manuscript preparation (e.g. some of the text in the manuscript does not match the presented trees; sequence assignment to countries in incorrect in phylogenetic trees [Figure 9]; please see below).
In my opinion, in the presented format, the manuscript is not acceptable for publication. A thorough revision to increase clarification and correct the multiple faults is imperative. English language editing is also necessary to correct several sentences (e.g., several written in past tense that should be in the present tense).
Specifically, I have the following comments:
· The Abstract should be reformulate in a way synthetize the relevant information.
· The collection of ticks is reported from five districts in Pakistan (Materials and Methods and Figure 1), and nucleic acid extraction and cDNA synthesis is reported from 20 R. microplus ticks representing each district. Nevertheless, in the results, only one sequence for cox and immunogenic sequences is presented (one accession number for each gene target). Are all the sequences from the same tick? From which region? What happen to the other sequences/ticks? All the information from the tick collection sites is lost in the presented Results.
· Immunogenic sequences description should be more detailed (with full name and gene abbreviation description for all possible ORF/target genes) to clarify their function and importance in anti-tick vaccine development.
· Reference for Mega X software is not the presented in reference 35. This reference is for Mega 11… The references should match the used software.
· Figure 2- cox phylogenetic tree: did you used nucleotide or aa sequences? It is not referred in the results (line 213-215) nor in the Figure legend.
· Figure 3, Figure 6, Figure 10 – For Figure 3 the text refers that the deduced aa sequence showed 100% identity (ID) with the homologous sequences of R. microplus from India and Brazil, but the tree shows that only the sequence from India presents 100% ID, the sequence from Brazil is not identical. Is the tree based in nucleotide sequences and incorrectly described as using aa sequences? The tree figures is not compatible with the text description. The same type of incompatible information is also presented for Figure 6 and Figure 10, since the presented trees show that the sequences are not 100% identical, as stated in the manuscript text.
· Figure 7 – the tick from Pakistan used for P0 sequence (60S acidic ribosomal protein – the full description of this sequence should be made) is not R. microplus??? The sequence species is identified in the tree as R. sanguineus… I big confusing or mistake was made here.
· Figure 9 – most sequences used are wrongly assigned to Netherlands. Only sequence ADR01308 is from Netherlands, all the others are from ticks collected in different countries, several from South Africa, but also from France, USA, China and Gambia.
· Table 2- Why the sequences/Accession numbers presented in Table 2 here not included in the phylogenetic analysis performed. e.g. The closer sequence included in Figure 10, in immunogenic protein Rm05Uy tree is from China, and in Table 2 a sequence from Brazil is presented…
· Some tick species names are not in italic and the use of abbreviations of the genera description is not homogeneously used for the first species of the referred genus in the same sentence and along the text (e.g. lines 223-225, 249-250, 261-264, 275-277).
Author Response
Reviewer comment:
The manuscript describes the analysis of immunogenic sequences of Rhipicephalus microplus ticks from Pakistan. This study aims to characterize these immunogenic sequences and is presented as the first preliminary report from R. microplus phylogenetic clade C from this region, with impact in the development of an anti-tick vaccine.
Author’s response:
Thank you for the revision of the MS.
Reviewer comment:
The manuscript data could be soundness it correctly presented, since it provides new data (and regarding the importance of tick control and related tick-borne diseases transmission). However, there are several important faults: data that is not clear, coherent, and/or is incorrectly presented. The results presentation should have been more care-fully addressed during manuscript preparation (e.g. some of the text in the manuscript does not match the presented trees; sequence assignment to countries in incorrect in phylogenetic trees [Figure 9]; please see below).
Author’s response:
Thank you for the suggestions, all the suggestions have been addressed in the revised MS. The phylogenetic tree (Figure 9) was modified as suggested.
Reviewer comment:
In my opinion, in the presented format, the manuscript is not acceptable for publication. A thorough revision to increase clarification and correct the multiple faults is imperative. English language editing is also necessary to correct several sentences (e.g., several written in past tense that should be in the present tense).
Author’s response:
Suggestions accepted as suggested.
Reviewer comment:
The Abstract should be reformulate in a way synthetize the relevant information.
Author’s response:
The abstract was modified for relevant information as suggested.
Reviewer comment:
The collection of ticks is reported from five districts in Pakistan (Materials and Methods and Figure 1), and nucleic acid extraction and cDNA synthesis is reported from 20 R. microplus ticks representing each district. Nevertheless, in the results, only one sequence for cox and immunogenic sequences is presented (one accession number for each gene target). Are all the sequences from the same tick? From which region? What happen to the other sequences/ticks? All the information from the tick collection sites is lost in the presented Results.
Author’s response:
All the obtained sequences of cox and immunogenic gene from each district were 100% identical, therefore a consensus sequence was obtained in the SeqMan which was used in the sequences and phylogenetic analysis and uploaded to GenBank.
Reviewer comment:
Immunogenic sequences description should be more detailed (with full name and gene abbreviation description for all possible ORF/target genes) to clarify their function and importance in anti-tick vaccine development.
Author’s response:
Thank you for the suggestion, details have been added to all the possible ORF/target genes as suggested.
Reviewer comment:
Reference for Mega X software is not the presented in reference 35. This reference is for Mega 11… The references should match the used software.
Author’s response:
Suggestion accepted and the original source/reference was added to the reference list (Reference 35).
Reviewer comment:
Figure 2- cox phylogenetic tree: did you used nucleotide or aa sequences? It is not referred in the results (lines 213-215) nor in the Figure legend.
Author’s response:
Phylogenetic tree for cox was constructed using nucleotide sequences. The result and figure 2 legend were modified as suggested.
Reviewer comment:
Figure 3, Figure 6, Figure 10 – For Figure 3 the text refers that the deduced aa sequence showed 100% identity (ID) with the homologous sequences of R. microplus from India and Brazil, but the tree shows that only the sequence from India presents 100% ID, the sequence from Brazil is not identical. Is the tree based in nucleotide sequences and incorrectly described as using aa sequences? The tree figures is not compatible with the text description. The same type of incompatible information is also presented for Figure 6 and Figure 10, since the presented trees show that the sequences are not 100% identical, as stated in the manuscript text.
Author’s response:
The phylogenetic trees were constructed using amino acid sequences. The Blastx identities of the Cystatin 2b (figure 3), and Rm05Uy (figure 10) were corrected in the MS as suggested. The phylogenetic trees for Ferritin 1 (figure 6) and Rm05Uy (figure 10) were reconstructed and changes were incorporated in the abstract, result, and discussion sections.
Reviewer comment:
Figure 7 – the tick from Pakistan used for P0 sequence (60S acidic ribosomal protein – the full description of this sequence should be made) is not R. microplus??? The sequence species is identified in the tree as R. sanguineus… I big confusing or mistake was made here.
Author’s response:
Thank you. The term P0 was described as 60S acidic ribosomal protein (P0) as suggested (paper reference: https://doi.org/10.1016/j.ttbdis.2015.04.007). The same sequence was amplified from the R. microplus. The phylogenetic tree was reconstructed in the MS as suggested (Figure 7).
Reviewer comment:
Figure 9 – most sequences used are wrongly assigned to Netherlands. Only sequence ADR01308 is from Netherlands, all the others are from ticks collected in different countries, several from South Africa, but also from France, USA, China and Gambia.
Author’s response:
Each accession number of the tree was searched and the phylogenetic tree was corrected as suggested (Figure 9) in the MS.
Reviewer comment:
Table 2- Why the sequences/Accession numbers presented in Table 2 here not included in the phylogenetic analysis performed. e.g. The closer sequence included in Figure 10, in immunogenic protein Rm05Uy tree is from China, and in Table 2 a sequence from Brazil is presented.
Author’s response:
In Table 2, the nucleotide sequence was used for the nucleotide polymorphism and associated nonsynonymous polymorphism however, the phylogenetic analysis was constructed using amino acid sequences. Rm05Uy sequence in table 2 and phylogenetic tree was modified as suggested.
Reviewer comment:
Some tick species names are not in italic and the use of abbreviations of the genera description is not homogeneously used for the first species of the referred genus in the same sentence and along the text (e.g. lines 223-225, 249-250, 261-264, 275-277).
Author’s response:
Thank you. All the names and abbreviations have been checked and corrected as suggested. The genus name was abbreviated after its first use following the same for different species within that genus.
Round 2
Reviewer 2 Report
Although some data was corrected in this review version of the manuscript, parts of the text in the manuscript still do not match the presented trees, the phylogenetic analysis of the immunogenic aa sequences is presented with overstatements regarding the inference of the evolutionary story of the referred genes, and English language editing is still necessary to correct and clarify several sentences.
In my opinion, the manuscript is still not acceptable for publication.
Specifically, I have the following comments:
· The Abstract was only slightly reformulated.
· The Phylogenetic analysis of the immunogenic aa sequences is described using overstatements. More care should be taken is these descriptions. The evolutionary history cannot be inferred (nor compared) using an analysis with such a diverse sequence data background. To assume monophyletic clades in these trees is highly controversial. To start the gene sequences are not DNA barcoding genes, the “sampling effort” and the used sequences at each tree is completely different (from the available sequences in databases; with different tick species and genus) and the used data will influence the resulting trees and bias the obtained phylogenies. So, all sentences describing the phylogenetic tree as “clustered in a monophyletic clade” should be reformulated to “cluster with the most similar available sequences…”. Monophyly estimates are influenced by tree-building methods, sampling effort, gene sequences and other methodological issues, and should not, in my opinion, be referred in these phylogenetic sequence data analysis (immunogenic aa sequences) with such a variable sequence data input in each target sequence analysis.
· Still in Phylogenetic analysis most descriptions state “In a phylogenetic analysis…” (Lines 227, 240, 253, 281, 294, 307, 319) and should be changed to “in the phylogenetic analysis…” or “In the presented/performed phylogenetic analysis…”.
· Immunogenic sequences description should be more detailed (with full name and gene abbreviation description for all possible ORF/target genes) in the introduction to clarify their function and why were chosen in this study/ importance in anti-tick vaccine development.
· Figure 10 – still the text refers that the deduced aa sequence showed 100% identity (ID) with the homologous sequences of R. microplus now from Uruguay (not China as in the previous version), but the tree show that the sequences are not 100% identical, as stated in the manuscript text.
· Still for some tick species names the use of abbreviations of the genera description is not homogeneously used for the first species of the referred genus in the same sentence and along the text.
So, in lines 224-226: should be “The sequence identity to the respective R. appendiculatus, R. sanguineus, R. haemaphysaloides, Haemaphysalis flava, H. longicornis, and I. scapularis…”;
in lines 238-240: should be “several other tick species including R. annulatus, R. haemaphysaloides, R. sanguineus, Hyalomma anatolicum, Dermacentor silvarum, D. variabilis, H. flava, H. longicornis, Amblyomma variegatum, and I. ricinus were 63.36-98.80%.”;
in lines 251-253 should be: “Sequence identities to the R. annulatus, R. sanguineus, D. silvarum, D. marginatus, D. variabilis, A. variegatum, H. longicornis, and I. scapularis were 89.24-99.26%.”;
in lines 263-265: should be “Maximum sequence identities with other ticks including R. sanguineus, Hyalomma anatolicum, Hy. asiaticum, R. haemaphysaloides, D. silvarum, D. andersoni, D. variabilis, A. americanum, A. maculatum, H. longicornis, H. flava, H. doenitzi, I. scapularis, and I. ricinus were 84.88-96.51%.”,
in lines 278-280: should be: “Sequence identities with the homologous sequences in R. haemaphysaloides, D. andersoni, Hy. asiaticum, D. nitens, D. silvarum, A. cajennense, A. variegatum, A. maculatum, H. longicornis, and I. scapularis were 75.4- 280 99.69%.).
· Line 19-20 : “…and the identification of tick-derived highly immunogenic sequences for the development of an anti-tick vaccine has been emerged as a successful alternate.” Suggestion, should be “and the identification of tick-derived highly immunogenic sequences for the development of an anti-tick vaccines has emerged as a successful alternate.”
· Line 47-48: Sentence “Mainly in tropical and subtropical countries where cattle popula- tions have been addressed at risk of ticks and tick-borne diseases, representing huge estimated economic impact” these sentence needs to be reformulated or linked to the previous sentence.
· Line 107: reference for ArcGIS is missing
· Line 123: “Furthermore, 1 µg/µl RNA was mixed with 1 µl of 100 µM oligo (dT)18,” – 18 in superscript should be removed, right?
· Line 163: “The final extension was kept at 72 °C for 7-10 min, 163 and then held the reaction at 4 °C.” change suggestion to: “The final extension was performed at 72 °C for 7-10 min, and then helded at 4 °C until further processing.”
· Lines 173-182: Why cloning before sequencing? Why not direct sequencing of the PCR products? I question if this approach might have induce the loss of some sequence variability…
· Lines 184-187: The changed sentence is confusing and needs to be rephrased for clarification.
· Line 189: Incorrect: “ The phylogenetic trees…”
· Lines 193-194: Not “An outgroup sequence..” since in several trees are presented with two/more than one outgroup sequences.
Author Response
Reviewer #2:
Reviewer comment:
Although some data was corrected in this review version of the manuscript, parts of the text in the manuscript still do not match the presented trees, the phylogenetic analysis of the immunogenic aa sequences is presented with overstatements regarding the inference of the evolutionary story of the referred genes, and English language editing is still necessary to correct and clarify several sentences.
In my opinion, the manuscript is still not acceptable for publication. Specifically, I have the following comments:
Author’s response:
Thank you, all the suggestions were accepted and changes have been incorporated in the MS.
Reviewer comment:
The Abstract was only slightly reformulated.
Author’s response:
The abstract was revised as suggested.
Reviewer comment:
The Phylogenetic analysis of the immunogenic aa sequences is described using overstatements. More care should be taken is these descriptions. The evolutionary history cannot be inferred (nor compared) using an analysis with such a diverse sequence data background. To assume monophyletic clades in these trees is highly controversial. To start the gene sequences are not DNA barcoding genes, the “sampling effort” and the used sequences at each tree is completely different (from the available sequences in databases; with different tick species and genus) and the used data will influence the resulting trees and bias the obtained phylogenies. So, all sentences describing the phylogenetic tree as “clustered in a monophyletic clade” should be reformulated to “cluster with the most similar available sequences…”. Monophyly estimates are influenced by tree-building methods, sampling effort, gene sequences and other methodological issues, and should not, in my opinion, be referred in these phylogenetic sequence data analysis (immunogenic aa sequences) with such a variable sequence data input in each target sequence analysis.
Author response:
All the analysis were repeated and the phylogenetic tress (Figure 3, 5, 6, 8, 10) have been reconstructed by adding more sequences. The description of each immunogenic protein phylogenetic tree was modified as suggested.
Reviewer comment:
Still in Phylogenetic analysis most descriptions state “In a phylogenetic analysis…” (Lines 227, 240, 253, 281, 294, 307, 319) and should be changed to “in the phylogenetic analysis…” or “In the presented/performed phylogenetic analysis…”.
Author’s response:
Changes have been incorporated as suggested.
Reviewer comment:
Immunogenic sequences description should be more detailed (with full name and gene abbreviation description for all possible ORF/target genes) in the introduction to clarify their function and why were chosen in this study/ importance in anti-tick vaccine development.
Author’s response:
Suggestion accepted as suggested.
Reviewer comment:
Figure 10 – still the text refers that the deduced aa sequence showed 100% identity (ID) with the homologous sequences of R. microplus now from Uruguay (not China as in the previous version), but the tree show that the sequences are not 100% identical, as stated in the manuscript text.
Author’s response:
Thank you. The phylogenetic tress (Figure 10) was reconstructed.
Reviewer comment:
Still for some tick species names the use of abbreviations of the genera description is not homogeneously used for the first species of the referred genus in the same sentence and along the text. So, in lines 224-226: should be “The sequence identity to the respective R. appendiculatus, R. sanguineus, R. haemaphysaloides, Haemaphysalis flava, H. longicornis, and I. scapularis…”;
Author’s response:
The sentence was modified as suggested.
Reviewer comment:
In lines 238-240: should be “several other tick species including R. annulatus, R. haemaphysaloides, R. sanguineus, Hyalomma anatolicum, Dermacentor silvarum, D. variabilis, H. flava, H. longicornis, Amblyomma variegatum, and I. ricinus were 63.36-98.80%.”;
Author’s response:
The sentence was modified as suggested.
Reviewer comment:
In lines 251-253 should be: “Sequence identities to the R. annulatus, R. sanguineus, D. silvarum, D. marginatus, D. variabilis, A. variegatum, H. longicornis, and I. scapularis were 89.24-99.26%.”;
Author’s response:
The sentence was modified as suggested.
Reviewer comment:
In lines 263-265: should be “Maximum sequence identities with other ticks including R. sanguineus, Hyalomma anatolicum, Hy. asiaticum, R. haemaphysaloides, D. silvarum, D. andersoni, D. variabilis, A. americanum, A. maculatum, H. longicornis, H. flava, H. doenitzi, I. scapularis, and I. ricinus were 84.88-96.51%.”,
Author’s response:
The sentence was modified as suggested.
Reviewer comment:
In lines 278-280: should be: “Sequence identities with the homologous sequences in R. haemaphysaloides, D. andersoni, Hy. asiaticum, D. nitens, D. silvarum, A. cajennense, A. variegatum, A. maculatum, H. longicornis, and I. scapularis were 75.4- 280 99.69%.).
Author’s response:
The sentence was modified as suggested.
Reviewer comment:
Line 19-20 : “…and the identification of tick-derived highly immunogenic sequences for the development of an anti-tick vaccine has been emerged as a successful alternate.” Suggestion, should be “and the identification of tick-derived highly immunogenic sequences for the development of an anti-tick vaccines has emerged as a successful alternate.”
Author’s response:
The sentence was modified as suggested.
Reviewer comment:
Line 47-48: Sentence “Mainly in tropical and subtropical countries where cattle popula- tions have been addressed at risk of ticks and tick-borne diseases, representing huge estimated economic impact” these sentence needs to be reformulated or linked to the previous sentence.
Author’s response:
The sentence was linked with the previous sentence as suggested.
Reviewer comment:
Line 107: reference for ArcGIS is missing
Author’s response:
Reference has been added.
Reviewer comment:
Line 123: “Furthermore, 1 µg/µl RNA was mixed with 1 µl of 100 µM oligo (dT)18,” – 18 in superscript should be removed, right?
Author’s response:
Suggestion accepted as suggested.
Reviewer comment:
Line 163: “The final extension was kept at 72 °C for 7-10 min, 163 and then held the reaction at 4 °C.” change suggestion to: “The final extension was performed at 72 °C for 7-10 min, and then helded at 4 °C until further processing.”
Author’s response:
Suggestion accepted. The sentence was changed to “The final extension was performed at 72 °C for 7-10 min, and then held at 4 °C until further processing.”
Reviewer comment:
Lines 173-182: Why cloning before sequencing? Why not direct sequencing of the PCR products? I question if this approach might have induced the loss of some sequence variability.
Author’s response:
Cloning was performed to get the complete gene encoding sequences (first few and last few nucleotide sequence which often lost in the sequencing after PCR). The accurate detail was necessary as these sequences can be used in the protein expression for anti-tick vaccine development.
Reviewer comment:
Lines 184-187: The changed sentence is confusing and needs to be rephrased for clarification.
Author’s response:
The sentence has been changed.
Reviewer comment:
Line 189: Incorrect: “The phylogenetic trees…”
Author’s response:
The correction was incorporated in the MS.
Reviewer comment:
Lines 193-194: Not “An outgroup sequence.” since in several trees are presented with two/more than one outgroup sequences.
Author’s response:
The corrections were incorporated in the MS.
Round 3
Reviewer 2 Report
Thank you for the changes made in the manuscript. I think that a more thouroght revision would benefit this data presentation, in a way to improve the scientific soundness and overall merit of the manuscript.
I still have some doubts regarding how the results are presented and about the final phylogenetic trees. Nevertheless, I think I have already addressed all the points in the previous reviews.
Author Response
We are greatly thankful for your suggestions. The whole manuscript was thoroughly read and corrections were incorporated according to the suggestions. The analysis for the amino acid sequences was performed by researching proteins sequences using BLASTp (protein-to-protein), and BLASTx (Nucleotide-to-protein) and the most similar available complete sequences were downloaded for inclusion in the phylogenetic analysis. The methodological framework for the phylogenetic trees was modified in accordance to the use of each parameter and clustering method in MEGA 11 .